# RETHINKING GRAPH LOTTERY TICKETS: GRAPH SPARSITY MATTERS

**Bo Hui[1], Da Yan[2], Xiaolong Ma[3], Wei-Shinn Ku[1]**
[1] Auburn University [2] The University of Alabama at Birmingham [3] Clemson University

## ABSTRACT

Lottery Ticket Hypothesis (LTH) claims the existence of a winning ticket (i.e., a properly pruned sub-network together with original weight initialization) that can achieve competitive performance to the original dense network. A recent work, called UGS, extended LTH to prune graph neural networks (GNNs) for effectively accelerating GNN inference. UGS simultaneously prunes the graph adjacency matrix and the model weights using the same masking mechanism, but since the roles of the graph adjacency matrix and the weight matrices are very different, we find that their sparsifications lead to different performance characteristics. Specifically, we find that **the performance of a sparsified GNN degrades significantly when the graph sparsity goes beyond a certain extent.** Therefore, we propose two techniques to improve GNN performance when the graph sparsity is high. First, UGS prunes the adjacency matrix using a loss formulation which, however, does not properly involve all elements of the adjacency matrix; in contrast, we add a new auxiliary loss head to better guide the edge pruning by involving the entire adjacency matrix. Second, by regarding unfavorable graph sparsification as adversarial data perturbations, we formulate the pruning process as a min-max optimization problem to gain the robustness of lottery tickets when the graph sparsity is high. We further investigate the question: Can the "retrainable" winning ticket of a GNN be also effective for graph transferring learning? We call it the transferable graph lottery ticket (GLT) hypothesis. Extensive experiments were conducted which demonstrate the superiority of our proposed sparsification method over UGS, and which empirically verified our transferable GLT hypothesis.

## 1 INTRODUCTION

Graph Neural Networks (GNNs) (Kipf & Welling, 2017; Hamilton et al., 2017) have demonstrated state-of-the-art performance on various graph-based learning tasks. However, large graph size and over-parameterized network layers are factors that limit the scalability of GNNs, causing high training cost, slow inference speed, and large memory consumption. Recently, Lottery Ticket Hypothesis (LTH) (Frankle & Carbin, 2019) claims that there exists properly pruned sub-networks together with original weight initialization that can be retrained to achieve comparable performance to the original large deep neural networks. LTH has recently been extended to GNNs by Chen et al. (2021b), which proposes a unified GNN sparsification (UGS) framework that simultaneously prunes the graph adjacency matrix and the model weights to accelerate GNN inference on large graphs. Specifically, two differentiable masks $\mathbf{m}_g$ and $\mathbf{m}_\theta$ are applied to the adjacency matrix $\mathbf{A}$ and the model weights $\boldsymbol{\Theta}$, respectively, during end-to-end training by element-wise product. After training, lowest-magnitude elements in $\mathbf{m}_g$ and $\mathbf{m}_\theta$ are set to zero w.r.t. pre-defined ratios $p_g$ and $p_\theta$, which basically eliminates low-scored edges and weights, respectively. The weight parameters are then rewound to their original initialization, and this pruning process is repeated until pre-defined sparsity levels are reached, i.e.,

$$\textbf{graph sparsity } 1 - \frac{\|m_g\|_0}{\|A\|_0} \geq s_g \quad \textbf{and} \quad \textbf{weight sparsity } 1 - \frac{\|m_\theta\|_0}{\|\Theta\|_0} \geq s_\theta,$$

where $\|.\|_0$ is the $L^0$ norm counting the number of non-zero elements.

Intuitively, UGS simply extends the basic parameter-masking algorithm of Frankle & Carbin (2019) for identifying winning tickets to also mask and remove graph edges. However, our empirical study

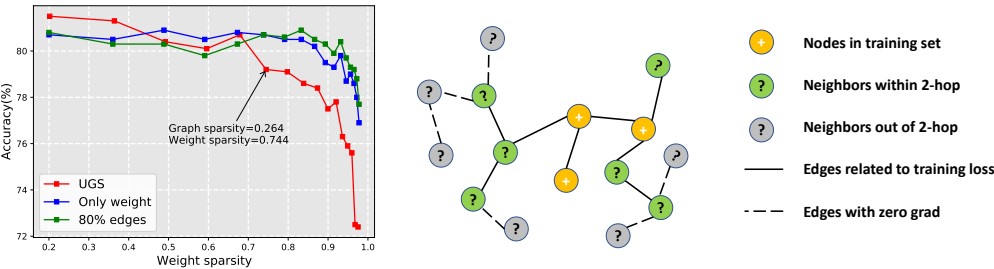

| (a) Performance Study of UGS Variants | (b) Types of Edges and Nodes in GNN Training |

Figure 1: UGS Analysis

finds that the performance of a sparsified GNN degrades significantly when the graph sparsity goes beyond a certain level, while it is relatively insensitive to weight sparsification. Specifically, we compare UGS with its two variants: (1) "Only weight," which does not conduct graph sparsification, and (2) "80% edges," which stops pruning edges as soon as the graph sparsity is increased to 20% or above. Figure 1(a) shows the performance comparison of UGS and the two variants on the Cora dataset (Chen et al., 2021b), where we can see that the accuracy of UGS (the red line) collapses when the graph sparsity becomes larger than 25%, while the two variants do not suffer from such a significant performance degradation since neither of them sparsifies the edges beyond 20%. Clearly, the performance of GNNs is vulnerable to graph sparsification: removing certain edges tends to undermine the underlying structure of the graph, hampering message passing along edges.

In this paper, we propose two techniques to improve GNN performance when the graph sparsity is high. The first technique is based on the observation that in UGS, only a fraction of the adjacency matrix elements (i.e., graph edges) are involved in loss calculation. As an illustration, consider the semi-supervised node classification task shown in Figure 1(b) where the nodes in yellow are labeled, and we assume that a GNN with 2 graph convolution layers is used so only nodes within two hops from the yellow nodes are involved in the training process, which are highlighted in green. Note that the gray edges in Figure 1(b) are not involved in loss calculation, i.e., no message passing happens along the dashed edges so their corresponding mask elements get zero gradients during backpropagation. On the Cora dataset, we find that around 50% edges are in such a situation, leaving the values of their corresponding mask elements unchanged throughout the entire training process. After checking the source code of UGS, we find that it initializes these mask elements by adding a random noise, so the ordering of the edge mask-scores is totally determined by this initial mask randomization rather than the graph topology. As a result, the removal of low-scored edges tends to be random in later iterations of UGS, causing performance collapse as some important "dashed" edges are removed.

To address this problem, we add a new auxiliary loss head to better guide the edge pruning by involving the entire adjacency matrix. Specifically, this loss head uses a novel loss function that measures the inter-class separateness of nodes with Wasserstein distance (WD). For each class, we calculate the WD between (1) the set of nodes that are predicted to be in the class and (2) the set of other nodes. By minimizing WD for all classes, we maximize the difference between the extracted node features of different classes. Now that this loss function involves all nodes, all elements in the graph mask $\mathbf{m}_g$ will now have gradient during backpropagation.

Our second technique is based on adversarial perturbation, which is widely used to improve the robustness of deep neural networks (Wong et al., 2020). To improve the robustness of the graph lottery tickets (i.e., the pruned GNN subnetworks) when graph sparsity is high, we regard unfavorable graph sparsification as an adversarial data perturbation and formulate the pruning process as a min-max optimization problem. Specifically, a minimizer seeks to update both the weight parameters $\Theta$ and its mask $\mathbf{m}_\theta$ against a maximizer that aims to perturb the graph mask $\mathbf{m}_g$. By performing projected gradient ascent on the graph mask, we are essentially using adversarial perturbations to significantly improve the robustness of our graph lottery tickets against the graph sparsity.

We further investigate the question: Can we use the obtained winning ticket of a GNN for graph transfer learning? Studying this problem is particularly interesting since the "retrainability" of a winning ticket (i.e., a pruned sub-network together with original weight initialization) on the same task is the most distinctive property of Lottery Ticket Hypothesis (LTH): many works (Liu

et al., 2019; Frankle & Carbin, 2019) have verified the importance of winning ticket initialization and stated that the winning ticket initialization might land in a region of the loss landscape that is particularly amenable to optimization. Here, we move one step further in the transfer learning setting, to investigate if the winning ticket identified on a source task also works on a target task. The answer is affirmative through our empirical study, and we call it the transferable graph LTH.

The contributions of this paper are summarized as follows:

- We design a new auxiliary loss function to guide the edges pruning for identifying graph lottery tickets. Our new auxiliary loss is able to address the issue of random edge pruning in UGS.

- We formalize our sparsification algorithm as a min-max optimization process to gain the robustness of lottery tickets to the graph sparsity.

- We empirically investigated and verified the transferable graph lottery ticket hypotheses.

## 2 PRELIMINARY

This section defines our notations, reviews the concept of GNNs, defines graph lottery tickets, and reports our empirical study of UGS's random edge pruning issue.

**Notations and GNN Formulation.** We consider an undirected graph $G = (\mathcal{V}, \mathcal{E})$ where $\mathcal{V}$ and $\mathcal{E}$ are the sets of nodes and edges of $G$, respectively. Each node $v_i \in \mathcal{V}$ has a feature vector $\mathbf{x}_i \in \mathbb{R}^F$, where $F$ is the number of node features. We use $\mathbf{X} \in \mathbb{R}^{N \times F}$ to denote the feature matrix of the whole graph $G$, where $N = |\mathcal{V}|$ is the number of nodes, and $\mathbf{X}$ stacks $\mathbf{x}_i$ of all nodes $v_i$ as its rows. Let $\mathbf{A} \in \mathbb{R}^{N \times N}$ be the adjacency matrix of $G$, i.e., $\mathbf{A}_{i,j} = 1$ if there is an edge $e_{i,j} \in \mathcal{E}$ between nodes $v_i$ and $v_j$, and $\mathbf{A}_{i,j} = 0$ otherwise. Graph Convolutional Networks (GCN) (Kipf & Welling, 2017) is the most popular and widely adopted GNN model. Without loss of generality, we consider a GCN with two graph convolution layers, which is formulated as follows:

$$\mathbf{Z} = \text{softmax}\left(\hat{\mathbf{A}}\,\sigma\left(\hat{\mathbf{A}}\mathbf{X}\mathbf{W}^{(0)}\right)\mathbf{W}^{(1)}\right) \in \mathbb{R}^{N \times C}. \tag{1}$$

Here, $\mathbf{W}^{(0)}$ is an input-to-hidden weight matrix and $\mathbf{W}^{(1)}$ is a hidden-to-output weight matrix. The softmax activation function is applied row-wise and $\sigma(\cdot) = \max(0, \cdot)$ is the ReLU activation function. The total number of classes is $C$. The adjacency matrix with self-connections $\tilde{\mathbf{A}} = \mathbf{A} + \mathbf{I}_N$ is normalized by $\hat{\mathbf{A}} = \tilde{\mathbf{D}}^{-\frac{1}{2}}\tilde{\mathbf{A}}\tilde{\mathbf{D}}^{-\frac{1}{2}}$ where $\tilde{\mathbf{D}}$ is the degree matrix of $\tilde{\mathbf{A}}$. For the semi-supervised node classification tasks, the cross-entropy error over labeled nodes is given by:

$$\mathcal{L}_0 = -\sum_{l \in \mathcal{Y}_L}\sum_{j=1}^{C} \mathbf{Y}_{lj}\log(\mathbf{Z}_{lj}), \tag{2}$$

where $\mathcal{Y}_L$ is the set of node indices for those nodes with labels, $\mathbf{Y}_l$ is the one-hot encoding (row) vector of node $v_l$'s label.

**Graph Lottery Tickets.** If a sub-network of a GNN with the original initialization trained on a sparsified graph has a comparable performance to the original GNN trained on the full graph in terms of test accuracy, then the GNN subnetwork along with the sparsified graph is defined as a graph lottery ticket (GLT) (Chen et al., 2021b). Different from the general LTH literature, GLT consists of three elements: (1) a sparsified graph obtained by pruning some edges in $G$, (2) a GNN sub-network and (3) the initialization of its learnable parameters.

Without loss of generality, let us consider a 2-layer GCN denoted by $f(\mathbf{X}, \boldsymbol{\Theta}_0)$ where $\boldsymbol{\Theta}_0 = \{\mathbf{W}_0^{(0)}, \mathbf{W}_0^{(1)}\}$ is the initial weight parameters. Then, the task is to find two masks $\mathbf{m}_g$ and $\mathbf{m}_\theta$ such that the sub-network $f(\mathbf{X}, \mathbf{m}_\theta \odot \boldsymbol{\Theta}_0)$ along with the sparsified graph $\{\mathbf{m}_g \odot \mathbf{A}, \mathbf{X}\}$ can be trained to a comparable accuracy as $f(\mathbf{X}, \boldsymbol{\Theta}_0)$.

**UGS Analysis.** LTH algorithms including UGS (Chen et al., 2021b) sort the elements in a mask by magnitude and "zero out" the smallest ones for sparsification. To find and remove the insignificant connections and weights in the graph and the GNN model, UGS updates two masks $\mathbf{m}_g$ and $\mathbf{m}_\theta$ by backpropagation. However, recall from Figure 1(b) that UGS suffers from the issue that only a fraction of edges in $G$ are related to the training loss as given by Eq (2) for semi-supervised node

classification. Table 1 shows (1) the number of nodes in the training v.s. entire dataset and (2) the percentage of edges with gradients during backpropagation for three graph datasets: Cora, Citeseer and PubMed. We can see that a significant percentage of edges (up to 91% as in PubMed) are not related to the training loss. We carefully examined the source code of UGS and found that the pruning process for these edges

Table 1: Statistic of Edges with Gradients

| Dataset | # of nodes (training/all) | Percentage of edges related to loss |
|---------|---------------------------|-------------------------------------|
| Cora | 140 / 2,708 | 51% (2,702 / 5,429) |
| Citeseer | 120 / 3,327 | 34% (1,593 / 4,732) |
| PubMed | 60 / 44,338 | 9% (3,712 / 44,338) |

tends to be random since each element in the graph mask is initialized by adding a random noise to its magnitude. As we have seen in Figure 1(a), the accuracy of UGS collapses when the graph sparsity becomes high. It is crucial to properly guide the mask learning for those edges not related to the training loss in Eq (2) to alleviate the performance degradation.

## 3 METHODOLOGY

**Auxiliary Loss Function.** Since $\mathcal{L}_0$ as given by Eq (2) does not involve all edges for backpropagation, we hereby design another auxiliary loss function $\mathcal{L}_1$ to better guide the edge pruning together with $\mathcal{L}_0$.

Given GNN output $\mathbf{Z}$ and a class $c$, we separate potential nodes of class $c$ and the other nodes by:

$$\mathbf{Z}^c = \{\mathbf{z}_i \in \text{rows}(\mathbf{Z}) \,|\, \text{argmax}(\mathbf{z}_i) = c\} \quad \text{and} \quad \mathbf{Z}^{\bar{c}} = \{\mathbf{z}_i \in \text{rows}(\mathbf{Z}) \,|\, \text{argmax}(\mathbf{z}_i) \neq c\}, \quad (3)$$

for $c \in \{1, 2, \cdots, C\}$. Since $\mathbf{z}_i$ of those $i \in \mathcal{Y}_L$ are properly guided by $\mathcal{L}_0$, we can use $\text{argmax}(\mathbf{z}_i) = c$ to filter out the potential nodes that belong to class $c$. For each class $c$, we then calculate the Wasserstein distance (WD) between $\mathbf{Z}^c$ and $\mathbf{Z}^{\bar{c}}$:

$$\text{WD}(\mathbf{Z}^c, \mathbf{Z}^{\bar{c}}) \quad = \inf_{\pi \sim \Pi(\mathbf{Z}^c, \mathbf{Z}^{\bar{c}})} \mathbb{E}_{(\mathbf{z_i}, \mathbf{z_j}) \sim \pi}[\|\mathbf{z_i} - \mathbf{z_j}\|], \quad (4)$$

where $\Pi(\mathbf{Z}^c, \mathbf{Z}^{\bar{c}})$ denotes the set of all joint distributions $\pi(\mathbf{z_i}, \mathbf{z_j})$ whose marginals are, respectively, $\mathbf{Z}^c$ and $\mathbf{Z}^{\bar{c}}$:

$$\Pi(\mathbf{Z}^c, \mathbf{Z}^{\bar{c}}) = \{\mathbf{P} : \mathbf{P1} = \mathbf{Z}^c, \mathbf{P}^T \mathbf{1} = \mathbf{Z}^{\bar{c}}\}. \quad (5)$$

Wasserstein distance (WD) (Villani, 2009) is widely used to measure the distance between two distributions, especially under the vanishing overlap of two distributions. Intuitively, it is the cost of the optimal plan for transporting one distribution to match another distribution: in Eq (5), $\Pi(\mathbf{Z}^c, \mathbf{Z}^{\bar{c}})$ is the set of valid transport plans, where $\mathbf{P}$ is a coupling matrix with each element $\mathbf{P}_{ij}$ representing how much probability mass from point $\mathbf{z}_i \in \mathbf{Z}^c$ is assigned to a point $\mathbf{z}_j \in \mathbf{Z}^{\bar{c}}$; and $\mathbf{1}$ is an all-one vector.

We consider $\mathbf{Z}^c$ and $\mathbf{Z}^{\bar{c}}$ as the distribution of class $c$ and other classes, respectively. Equation (4) can be approximated by a smooth convex optimization problem with an entropic regularization term:

$$\min_{\mathbf{P}} \langle \mathbf{D}, \mathbf{P} \rangle - \varepsilon \sum_{i,j} \mathbf{P}_{ij} \log(\mathbf{P}_{ij}), \quad (6)$$

where $\mathbf{D}$ is the pairwise Euclidean distance matrix between points of $\mathbf{Z}^c$ and $\mathbf{Z}^{\bar{c}}$, and $\langle \cdot, \cdot \rangle$ is the Frobenius inner product. Minimizing Eq (6) is solved using Sinkhorn iterations (Cuturi, 2013), which form a sequence of linear operations straightforward for backpropagation.

Ideally, we want the node representations in one class to be very different from those in other classes so that the classes can be easily distinguished. By maximizing $\text{WD}(\mathbf{Z}^c, \mathbf{Z}^{\bar{c}})$ for all classes $c$, we maximize the difference between GNN outputs for different classes. Formally, the auxiliary loss function to minimize is given by:

$$\mathcal{L}_1 = -\sum_{c \in \mathcal{C}} \text{WD}(\mathbf{Z}^c, \mathbf{Z}^{\bar{c}}). \quad (7)$$

To explore the relationship between unsupervised loss $\mathcal{L}_1$ with the supervised loss $\mathcal{L}_0$ (i.e., Eq (2)), we train a 2-layer GCN on Cora with $\mathcal{L}_0$ as the only loss function, and Figure 2 plots the value of $\mathcal{L}_0$ as well as the corresponding value of $\sum_{c \in \mathcal{C}} \text{WD}(\mathbf{Z}^c, \mathbf{Z}^{\bar{c}})$ (abbr. WD) during the training process. We can see that as training loss $\mathcal{L}_0$ decreases, the value of WD increases. The curves of both $\mathcal{L}_0$ and WD

start to bounce back after 100 epochs. Therefore, maximizing WD is consistent with improving classification accuracy. Note that $\sum_{c \in \mathcal{C}} \text{WD}(\mathbf{Z}^c, \mathbf{Z}^{\bar{c}})$ is unsupervised and related to all nodes in the training set. It is natural to use this new metric to guide the pruning of edges that are not directly related to $\mathcal{L}_0$: if masking an edge decreases WD, this edge tends to be important and its removal will decrease the test accuracy of node classification.

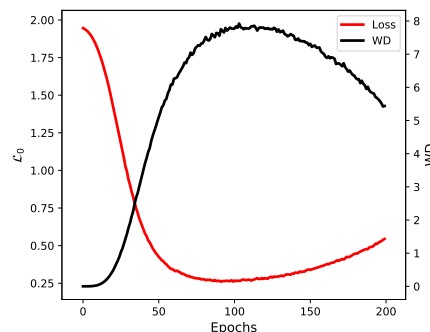

Our sparsification method uses a semi-supervised loss that combines $\mathcal{L}_0$ and $\mathcal{L}_1$ (or, WD): $\mathcal{L} = \mathcal{L}_0 + \lambda\mathcal{L}_1$.

**Sparsification.** To prune the edges and weights while retaining robustness, we formulate GNN sparsification as a min-max problem given by:

$$\min_{\boldsymbol{\Theta}, \mathbf{m}_\theta} \max_{\mathbf{m}_g} \ (\mathcal{L}_0 + \lambda\mathcal{L}_1), \qquad (8)$$

Figure 2: Supervised Loss $\mathcal{L}_0$ v.s. WD. Note that as training loss $\mathcal{L}_0$ decreases, the value of WD increases. Both $\mathcal{L}_0$ and WD start to bounce back after 100 epochs.

where the outer minimization seeks to train the model weights and their masks to minimize the classification error, and the inner maximization aims to adversarially perturb the graph mask.

Let us use $\mathcal{L}(\boldsymbol{\Theta}, \mathbf{m}_\theta, \mathbf{m}_g)$ to denote $(\mathcal{L}_0 + \lambda\mathcal{L}_1)$, then the optimization problem in Eq (8) can be solved by alternating the following two steps as specified by (i) Eq (9) and (ii) Eqs (10) & (11).

Specifically, the inner maximization perturbs the graph mask at $(t+1)^{\text{th}}$ iteration as follows:

$$\mathbf{m}_g^{(t+1)} \leftarrow \mathcal{P}_{[0,1]^{N \times N}} \left[ \mathbf{m}_g^{(t)} + \eta_1 \nabla_{\mathbf{m}_g} \mathcal{L}(\boldsymbol{\Theta}^{(t)}, \mathbf{m}_\theta^{(t)}, \mathbf{m}_g^{(t)}) \right], \qquad (9)$$

where $\mathcal{P}_{\mathcal{C}}[\mathbf{m}] = \arg\min_{\mathbf{m}' \in \mathcal{C}} \|\mathbf{m}' - \mathbf{m}\|_F^2$ is a differentiable projection operation that projects the mask values onto $[0, 1]$, and $\eta_1 > 0$ is a proper learning rate. While the second step for outer minimization is given by:

$$\mathbf{m}_\theta^{(t+1)} \leftarrow \mathbf{m}_\theta^{(t)} - \eta_2 \left( \nabla_{\mathbf{m}_\theta} \mathcal{L}(\boldsymbol{\Theta}^{(t)}, \mathbf{m}_\theta^{(t)}, \mathbf{m}_g^{(t+1)}) + \alpha \frac{d \left[ \mathbf{m}_g^{(t+1)} \right]^T}{d\mathbf{m}_\theta^{(t)}} \nabla_{\mathbf{m}_g} \mathcal{L}(\boldsymbol{\Theta}^{(t)}, \mathbf{m}_\theta^{(t)}, \mathbf{m}_g^{(t+1)}) \right),$$
$$(10)$$

$$\boldsymbol{\Theta}^{(t+1)} \leftarrow \boldsymbol{\Theta}^{(t)} - \eta_2 \left( \nabla_{\boldsymbol{\Theta}} \mathcal{L}(\boldsymbol{\Theta}^{(t)}, \mathbf{m}_\theta^{(t)}, \mathbf{m}_g^{(t+1)}) + \alpha \frac{d \left[ \mathbf{m}_g^{(t+1)} \right]^T}{d\boldsymbol{\Theta}^{(t)}} \nabla_{\mathbf{m}_g} \mathcal{L}(\boldsymbol{\Theta}^{(t)}, \mathbf{m}_\theta^{(t)}, \mathbf{m}_g^{(t+1)}) \right),$$
$$(11)$$

where the last term is an implicit gradient obtained by chain rule over $\mathbf{m}_g^{(t+1)}$ as given by Eq (9).

---

**Algorithm 1** Iterative pruning process

---

**Input**: Initial masks $\mathbf{m}_g = \mathbf{A}$, $\mathbf{m}_\theta = \mathbf{1} \in \mathbb{R}^{||\boldsymbol{\Theta}||}$
**Output**: Sparsified masks $\mathbf{m}_g$ and $\mathbf{m}_\theta$

1: **while** $1 - \frac{\|\mathbf{m}_g\|_0}{\|\mathbf{A}\|_0} < s_g$ and $1 - \frac{\|\mathbf{m}_\theta\|_0}{\|\boldsymbol{\Theta}\|_0} < s_\theta$ **do**
2:     $\mathbf{m}_g^{(0)} = \mathbf{m}_g, \mathbf{m}_\theta^{(0)} = \mathbf{m}_\theta, \boldsymbol{\Theta}^{(0)} = \{\mathbf{W}_0^{(0)}, \mathbf{W}_0^{(1)}\}$
3:     **for** iteration $t = 0, 1, 2, \cdots, T-1$ **do**
4:         Compute $\mathbf{m}_g^{(t+1)}$ with Eq (9)
5:         Compute $\mathbf{m}_\theta^{(t+1)}$ with Eq (10)
6:         Compute $\boldsymbol{\Theta}^{(t+1)}$ with Eq (11)
7:     **end for**
8:     $\mathbf{m}_g = \mathbf{m}_g^{(T-1)}, \mathbf{m}_\theta = \mathbf{m}_\theta^{(T-1)}$
9:     Set $p_g\%$ of the lowest-scored values in $\mathbf{m}_g$ to 0 and set the others to 1
10:    Set $p_\theta\%$ of the lowest-scored values in $\mathbf{m}_\theta$ to 0 and set the others to 1
11: **end while**

---

We follow UGS's framework to repeatedly train, prune, and reset a GNN with initial weights after each round until the desired sparsity levels $s_g$ and $s_\theta$ are reached. Algorithm 1 illustrates our iterative

pruning process. At each round once the model is trained and masks are computed, we set $p_g$ fraction of the lowest-scored values in $\mathbf{m}_g$ to 0 and set the others to 1. Likewise, $(1 - p_\theta)$ fraction of the weights in $\Theta$ will survive to the next round. Finally, we retrain the obtained sub-network using only $\mathcal{L}_0$ to report the test accuracy.

## 4 TRANSFER LEARNING WITH GRAPH LOTTERY TICKETS

This section explores whether a graph lottery ticket (GLT) can be used for graph transfer learning. Specifically, we ask two questions: (1) Can we use the graph winning ticket of a GNN for transfer learning? Here, the graph winning ticket is associated with the originally initialized weight parameters. (2) Can we use the post-trained winning ticket of a GNN for transfer learning? Here, "post-trained" means that the winning ticket directly uses the trained weight parameters on the source task to bootstrap the training on the target task.

To answer these two questions, we consider the node classification tasks on two citation networks arXiv and MAG (Hu et al., 2020). In both tasks, each node represents an author and is associated with a 128-d feature vector obtained by averaging the word embeddings of papers that are published by the author. We train a 3-layer GCN model on arXiv and transfer the model to MAG by replacing the last dense layer. Since the graphs of arXiv and MAG are different, we only transfer $\mathbf{m}_\theta$ and $\Theta$ from arXiv, and use the original graph of MAG without sparsification. Figure 3 shows the training accuracy curves on MAG with winning tickets obtained from arXiv. Here, "Post-trained GLT" (red line) starts training with the pretrained winning ticket on arXiv, while "Re-initialized GLT" (green line) starts training with the winning ticket re-initialized using original initial weight parameters. We can see that the sub-network obtained from arXiv transfers well on MAG in both settings, so the answers to our two questions are affirmative. Also, the blue (resp. black) line starts training with the post-trained GNN from arXiv without any edge

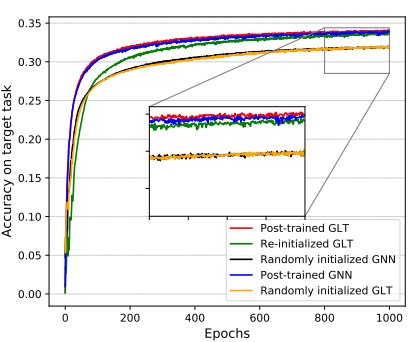

Figure 3: Training Curves on MAG. Notice that the post-trained winning ticket achieves the similar performance as the post-trained GNN. Also, the winning ticket achieves even better performance than a randomly initialized GNN.

or weight pruning; while the yellow line starts training with the winning ticket but with randomly initialized weight parameters. We can observe that (1) the post-trained winning ticket (red) achieves the same performance as the post-trained GNN (blue), showing that the transferred performance is not sensitive to weight pruning; and that (2) the winning ticket identified on the source task (green) achieves even better performance on the target task than a randomly initialized GNN (black). Note that gradient descent could get trapped in undesirable stationary points if it starts from arbitrary points, and our GLT identified from the source task allows the initialization to land in a region of the loss landscape that is particularly amenable to optimization (Ganea et al., 2018).

## 5 EXPERIMENT

**Experimental Setups.** Following Chen et al. (2021b), we evaluate our sparsification method with three popular GNN models: GCN (Kipf & Welling, 2017), GAT (Velickovic et al., 2018) and GIN (Xu et al., 2019), on three widely used graph datasets from Chen et al. (2021b) (Cora, Citeseer and PubMed) and two OGB datasets from Hu et al. (2020) (Arxiv and MAG) for semi-supervised node classification. We compare our method with two baselines: UGS (Chen et al., 2021b) and random pruning (Chen et al., 2021b). For fair comparison, we follow UGS to use the default setting: $p_g = 5$ and $p_\theta = 20$ unless otherwise stated. The value of $\lambda$ is configured as $0.1$ by default. More details on the dataset statistics and model configurations can be found in the appendix.

**Evaluation of Our Sparsification Method.** Figures 4, 5 and 6 compare our method with UGS and random pruning in terms of the test accuracy on three 2-layer GNN models: GCN, GIN and GAT, respectively. In each figure, the first (resp. second) row shows how the test accuracy changes with graph sparsity (resp. weight sparsity) during the iterative pruning process, on the three datasets: Cora, Citeseer and PubMed. We can observe that the accuracy of our method (green line) is much higher than UGS (yellow line) when the graph sparsity is high and when the weight sparsity is high.

Table 2: Pair-wise comparison

| Cora (17 iterations) | | | | | | | | | |
|---|---|---|---|---|---|---|---|---|---|
| Model | | GCN | | | GIN | | | GAT | |
| Methods/Metrics | GS(%) | WS(%) | Accuracy(%) | GS(%) | WS(%) | Accuracy(%) | GS(%) | WS(%) | Accuracy(%) |
| UGS • | 56.28 | 97.52 | 72.5 | 57.62 | 97.25 | 69.6 | 57.42 | 97.38 | 78.7 |
| Our ★ | **58.02** | **97.80** | **75.1** | **57.88** | **98.26** | **72.8** | **58.12** | **97.89** | **80.0** |
| Citeseer (17 iterations) | | | | | | | | | |
| Model | | GCN | | | GIN | | | GAT | |
| Methods/Metrics | GS(%) | WS(%) | Accuracy(%) | GS(%) | WS(%) | Accuracy(%) | GS(%) | WS(%) | Accuracy(%) |
| UGS • | 54.33 | 96.53 | 66.7 | **56.05** | 96.58 | 63.2 | 54.99 | 97.21 | 70.1 |
| Our ★ | **55.78** | **97.38** | **70.3** | 55.95 | **97.25** | **66.8** | **56.03** | **97.30** | **70.6** |
| PubMed (17 iterations) | | | | | | | | | |
| Model | | GCN | | | GIN | | | GAT | |
| Methods/Metrics | GS(%) | WS(%) | Accuracy(%) | GS(%) | WS(%) | Accuracy(%) | GS(%) | WS(%) | Accuracy(%) |
| UGS • | 58.21 | 97.76 | 77.7 | **58.23** | 97.19 | 76.3 | 57.64 | 97.32 | 78.8 |
| Our ★ | **59.43** | **97.81** | **79.5** | 58.13 | **97.76** | **77.2** | **58.22** | **98.18** | **80.0** |

To clearly see this accuracy difference, we show the test accuracy, graph sparsity (GS) and weight sparsity (WS) after the 17th pruning iteration in Table 2 for both UGS and our method. Note that each graph sparsity is paired with a weight sparsity. We use • and ★ to represent UGS and our method, respectively. Given the same sparcification iteration number, the sub-networks found by our method have a higher accuracy than those found by UGS. For example, when GS = 56.28% and WS = 97.52%, the accuracy of GCN on Cora is 72.5%, whereas our method improves the accuracy to 75.1% with a comparable setting: GS = 57.95% and WS = 97.31%. Note that although we use the same $p_g$ and $p_\theta$ for both our method and UGS, their sparsity values in the same pruning iteration are comparable but not exactly equal. This is because multiple mask entries can have the same value, and they span across the pruning threshold $p_g$ or $p_\theta$; we prune them all in an iteration such that slightly more than $p_g$ fraction of edges or $p_\theta$ fraction of weights may be removed. The higher accuracy of our method over UGS demonstrates that our approach finds lottery tickets that are more robust against the sparsity, and better prevents a performance collapse when the graph sparsity is high. In addition, we use • and ★ to indicate the last GLT with an accuracy higher than the original model in the sparsification process of UGS and our method, respectively. We can observe that the GLTs identified by our method have higher sparsity than those located by UGS. It further verifies that our method can improve the robustness of GLT against the sparsity.

Note that in Figure 4, the third subfigure, the accuracy of our method (green line) is very stable even when graph sparsity is high. This is because PubMed is a dense graph where some edges can result from noise during data collection, so graph sparsification functions as denoising, which retains or even improves model accuracy by alleviating the oversmoothing problem of GCN.

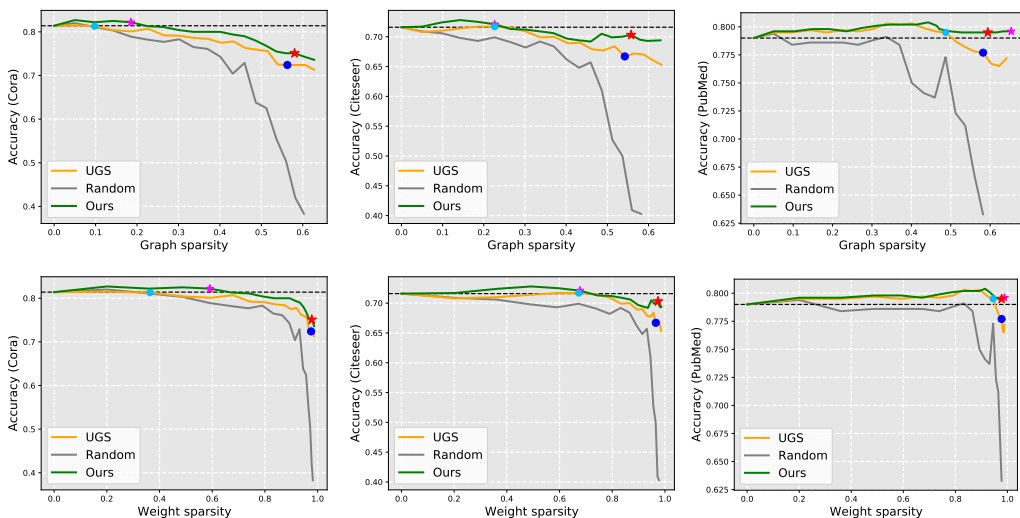

Figure 4: Performance on GCN. Marker • and ★ denote the spasified GCN (after 17 iterations) with UGS and our method, respectively. Marker • and ★ indicate the last GLT that reaches higher accuracy than the original model in the sparsification process of UGS and our method, respectively.

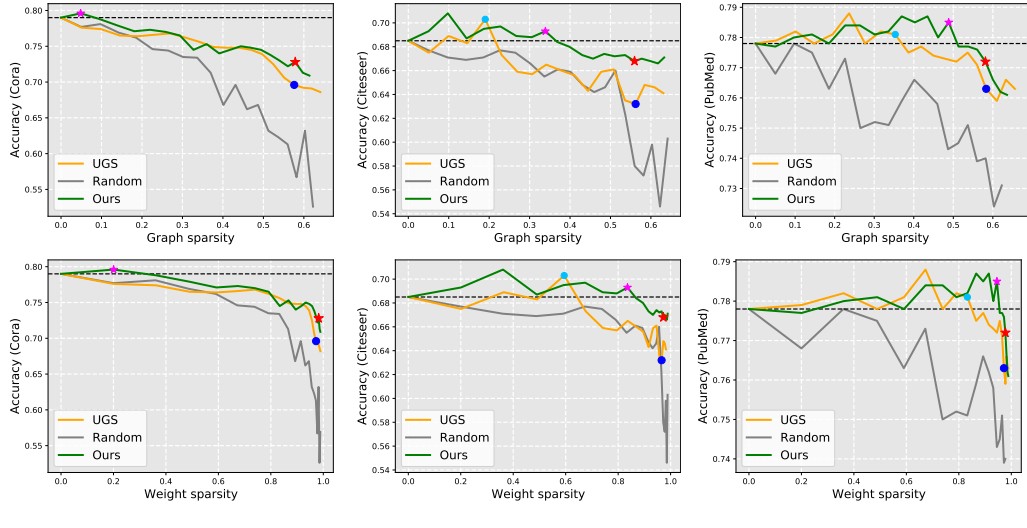

Figure 5: Performance on GIN

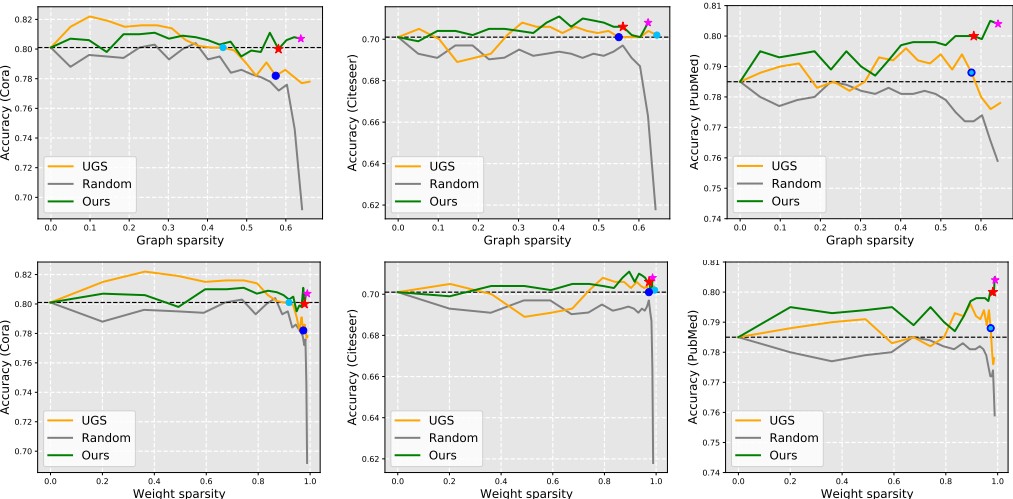

Figure 6: Performance on GAT

Also note from Figure 6 that the performance of GAT is less sensitive to the sparsity as compared with GCN and GIN. This is because the attention mechanism in GAT leads to many more parameters than the other two models, allowing for pruning more parameters; moreover, GAT itself is able to identify the importance of edges through the attention mechanism, which better guides the edge pruning.

**Ablation Study.** To verify the effectiveness of each design component in our sparsification algorithm, we compare our method with several variants. First, we remove the Wasserstein-distance based auxiliary loss head by setting $\lambda = 0$ during sparsification. As shown in Figure 7, removing the auxiliary loss function leads to performance degradation. However, there

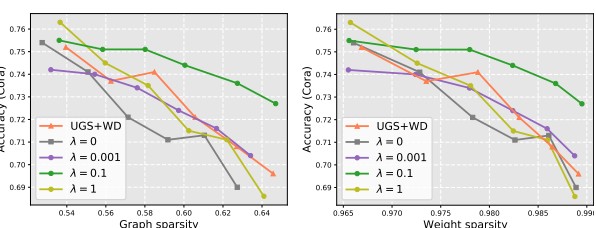

Figure 7: Ablation study

exists a tradeoff to weight the importance of $\mathcal{L}_1$. For example, if we increase $\lambda$ from 0.1 to 1, the performance backfires especially when the sparsity is high. We further investigate if our adversarial perturbation strategy can improve the robustness of the pruned GNNs, by replacing our min-max optimization process with the algorithm in UGS. Specifically, the variant "UGS+WD" uses the sparsification algorithm in UGS to minimize $\mathcal{L}(\theta, m_\theta, m_g)$. Compared with this variant, our method

can alleviate the performance degradation when graph sparsity is high, which verifies that adversarial perturbation improves robustness.

**Pairwise Sparsity.** In Figure 8, we investigate the accuracy of the pruned GNNs after 10 iterations with different combinations of $p_g$ and $p_\theta$. Note that larger $p_g$ and $p_\theta$ result in higher sparsity of the adjacent matrix and the weight matrices, respectively. We can see that the performance of the pruned GNN is more sensitive to the graph sparsity than the weight sparsity. It further verifies the necessity of finding robust GLTs when the graph sparsity is high. Since the pruned GNN is less sensitive to the weight sparsity, we can obtain GLTs with higher overall sparsity by setting $p_\theta$ to be larger $p_g$. At the same time, a larger $p_g$ can result in the pruning of more noisy and unimportant edges, which may help prevent GCN oversmoothing. Therefore, a tradeoff exists when choosing a proper combination of $p_g$ and $p_\theta$. While users

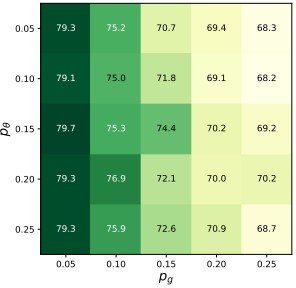

Figure 8: Varying $p_g$ and $p_\theta$

may try different combinations of $p_g$ and $p_\theta$ to find a proper combination, our approach is superior to UGS in all combinations of $p_g$ and $p_\theta$, so should be a better algorithm choice.

## 6 RELATED WORK

In the last few years, a series of GNN models (Kipf & Welling, 2017; Velickovic et al., 2018; Xu et al., 2019; Hamilton et al., 2017; Ying et al., 2018; Klicpera et al., 2019; Hui et al., 2020; Tong et al., 2020; Yun et al., 2019) have been designed to utilize the graph structure in spatial domain and spectral domain for effective feature extraction. However, the computational cost of GNNs on large-scale graphs can be very high especially when the graph is dense with many edges. Many models have been proposed to reduce this computational cost, such as FastGCN (Chen et al., 2018) and SGC (Wu et al., 2019). However, their accuracy tends to be worse than GCN, while the speedup achieved is still limited. So, it is desirable to develop a pruning method to sparsify both the edges and the weight parameters for any given GNN model (including FastGCN and SGC) without accuracy compromise.

The lottery ticket hypothesis is first proposed in Frankle & Carbin (2019) to find an identically initialized sub-network that can be trained to achieve a similar performance as the original neural network. Many works (Frankle et al., 2019; 2020; Diffenderfer & Kailkhura, 2021; Malach et al., 2020; Ma et al., 2021b; Savarese et al., 2020; Zhang et al., 2021a;b; Zhou et al., 2019; Chen et al., 2021c; Su et al., 2020; Chen et al., 2022) have verified that the identified winning tickets are retrainable to reduce the high memory cost and long inference time of the original neural networks. LTH has been extended to different kinds of neural networks such as GANs (Chen et al., 2021e;a; Kalibhat et al., 2021), Transformers (Brix et al., 2020; Prasanna et al., 2020; Chen et al., 2020; Behnke & Heafield, 2020) and even deep reinforcement learning models (Vischer et al., 2021; Yu et al., 2020). It has also been studied on GNNs by UGS (Chen et al., 2021b).

Despite the competitive performance of winning tickets, the iterative train-prune-retrain process and even the subsequent fine-tuning (Liu et al., 2021) are costly to compute. To address this issue, PrAC (Zhang et al., 2021c) proposes to find lottery tickets efficiently by using a selected subset of data rather than using the full training set. Other works (Wang et al., 2020; Tanaka et al., 2020) attempt to find winning tickets at initialization without training. The new trend of dynamic sparse training shows that any random initialized sparse neural networks can achieves comparable accuracy to the dense neural networks (Ye et al., 2020; Evci et al., 2020; Hou et al., 2022; Ma et al., 2021a; Yuan et al., 2021). Customized hardware architectures have also been used to accelerate sparse training (Goli & Aamodt, 2020; Raihan & Aamodt, 2020). The Elastic Lottery Ticket Hypothesis Chen et al. (2021d) studies the transferability of winning tickets between different network architectures within the same design family. For object recognition, Mehta (2019) shows that winning tickets from VGG19 do not transfer well to down-stream tasks. In the natural image domain, Morcos et al. (2019) finds that the winning ticket initializations generalize across a variety of vision benchmark datasets.

## 7 CONCLUSION

In this paper, we empirically observe that the performance of GLTs collapses when the graph sparsity is high. We design a new auxiliary loss to address this limitation. Moreover, we formulate the GNN sparsification process as a min-max optimization problem which adopts adversarial perturbation to improve the robustness of graph lottery tickets. Our experiments verified the superiority of our method over UGS. We further empirically studied transfer learning with GLTs and confirmed the transferability that gives rise to the transferable graph lottery ticket hypotheses.

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

# A   APPENDIX

## A.1   DATASET AND CONFIGURATION

The statistics of benchmark datasets in the experiment are summarized in Table 3. For three citation networks: Cora, CiteSeer and PubMed, the feature of each node corresponds to the bag-of-words representation of the document. For arXiv and MAG, each node is associated with a 128-dimensional feature vector obtained by averaging the word embeddings of papers. Three models (GCN, GIN and GAT) are configured with the default parameters in their corresponding source code, including number of layers, dimension of hidden state and learning rate. In our sparsification process, the value of $\eta_1$, $\eta_2$ and $\alpha$ are configured as 1e-2, 1e-2 and 1e-1 by default, respectively. In each pruning round, the number of epochs to update masks is configured as the default number of epochs in the training process of the original GNN model.

Table 3: Statistics of datasets

| Dataset | #Nodes | #Edges | Average Deg. | Split ratio | Features | Classes |
|---|---|---|---|---|---|---|
| Cora | 2,708 | 5,429 | 3.88 | 120/500/1000 | 1,433 | 7 |
| Citeseer | 3,327 | 4,732 | 2.84 | 140/500/1000 | 3,703 | 6 |
| PubMed | 19,717 | 44,338 | 4.50 | 60/500/1000 | 500 | 3 |
| arXiv | 169,343 | 1,166,243 | 13.7 | 54%18%/28% | 128 | 40 |
| MAG | 1,939,743 | 21,111,007 | 21.7 | 85%/9%/6% | 128 | 349 |

Table 4: GLT comparison

| Cora | | | | | | | | | |
|---|---|---|---|---|---|---|---|---|---|
| Model | | GCN | | | GIN | | | GAT | |
| Methods/Metrics | GS(%) | WS(%) | Accuracy(%) | GS(%) | WS(%) | Accuracy(%) | GS(%) | WS(%) | Accuracy(%) |
| UGS • | 9.76 | 36.42 | 81.3 | – | – | – | 43.96 | 91.88 | 80.1 |
| Ours ★ | **18.52** | **59.12** | **82.2** | **4.86** | **20.07** | **79.6** | **63.82** | **98.93** | **80.7** |

| Citeseer | | | | | | | | | |
|---|---|---|---|---|---|---|---|---|---|
| Model | | GCN | | | GIN | | | GAT | |
| Methods/Metrics | GS(%) | WS(%) | Accuracy(%) | GS(%) | WS(%) | Accuracy(%) | GS(%) | WS(%) | Accuracy(%) |
| UGS • | **22.65** | 67.34 | 71.7 | 19.03 | 59.43 | **70.3** | **64.48** | **99.41** | 70.2 |
| Ours ★ | 22.63 | **67.76** | **72.1** | **33.89** | **83.57** | 69.3 | 62.29 | 98.63 | **70.8** |

| PubMed | | | | | | | | | |
|---|---|---|---|---|---|---|---|---|---|
| Model | | GCN | | | GIN | | | GAT | |
| Methods/Metrics | GS(%) | WS(%) | Accuracy(%) | GS(%) | WS(%) | Accuracy(%) | GS(%) | WS(%) | Accuracy(%) |
| UGS • | 48.69 | 94.53 | 79.5 | 35.32 | 83.27 | 78.1 | 57.64 | 97.32 | 78.8 |
| Ours ★ | **65.35** | **98.91** | **79.6** | **48.84** | **94.52** | **78.5** | **64.26** | **99.07** | **80.4** |

## A.2 GLT comparison

In Table 4, we show the test accuracy, graph sparsity (GS) and weight sparsity (WS) of **the last GLT that reaches higher accuracy than the original model** in the sparsification process of UGS and our method. Note that GLT of GIN cannot be found on Cora in the sparsification process. We can observe that the last GLT identified by our method either has higher sparsity or achieves higher accuracy with comparable sparsity.

In Table 5, we evaluate the test accuracy of sparsified GCN on two large-scale graph datasets: arXiv and MAG. We can observe that our method can result in higher accuracy than UGS with comparable graph sparsity and weight sparsity after 5, 15, and 20 iterations. It further verifies the effectiveness of the proposed method on large-scale datasets.

Table 5: Performance of GCN on large-scale graph datasets

| | | | | | | | | | |
|---|---|---|---|---|---|---|---|---|---|
| ogbn-arxiv | | | | | | | | | |
| Iterations | 5 | | | 15 | | | 20 | | |
| Methods/Metrics | GS(%) | WS(%) | Accuracy(%) | GS(%) | WS(%) | Accuracy(%) | GS(%) | WS(%) | Accuracy(%) |
| UGS | 22.65 | 67.23 | 71.92 | 53.69 | 96.49 | 71.15 | 64.27 | 98.86 | 69.87 |
| Ours | 23.51 | 67.24 | 71.94 | 52.81 | 96.52 | 71.89 | 63.82 | 98.93 | 70.79 |
| ogbn-mag | | | | | | | | | |
| Iterations | 5 | | | 15 | | | 20 | | |
| Methods/Metrics | GS(%) | WS(%) | Accuracy(%) | GS(%) | WS(%) | Accuracy(%) | GS(%) | WS(%) | Accuracy(%) |
| UGS | 23.66 | 67.31 | 33.43 | 55.99 | 96.50 | 30.71 | 67.42 | 98.77 | 30.12 |
| Ours | 22.65 | 67.31 | 34.11 | 54.81 | 96.49 | 32.23 | 63.59 | 98.85 | 31.76 |

## A.3 Performance standard and inference time

To combat randomness, we visualize the standard deviation of accuracy on Cora with a shaded color. Following UGS, we also show the inference MACs (multiply–accumulate operations) of pruned models in the sparsification process. We can see that our method can significantly reduce the inference cost of GNNs with less performance drop.

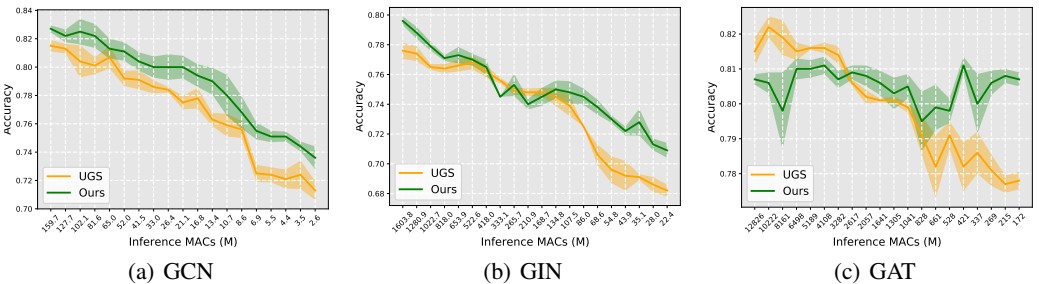

(a) GCN        (b) GIN        (c) GAT

Figure 9: Performance over inference MACs of GCN, GIN, and GAT on Cora

## A.4 Transfer Learning with GLTs

To verify the transferable graph lottery ticket hypothesis, we identify the GLT on arXiv and investigate the transfer ratio Kooverjee et al. (2022) of this GLT on MAG. Since the number of classes on arXiv

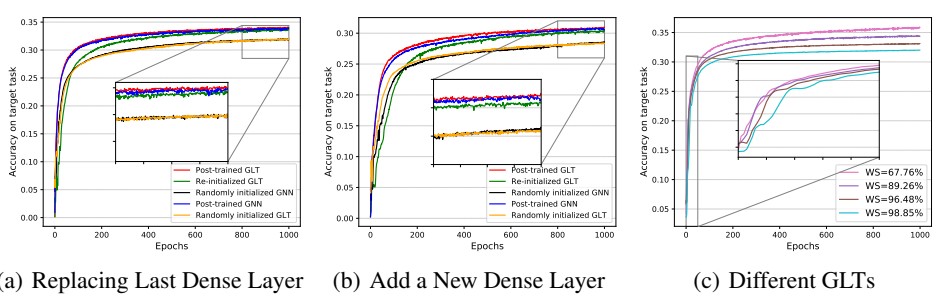

(a) Replacing Last Dense Layer    (b) Add a New Dense Layer    (c) Different GLTs

Figure 10: Transfer Learning from Arxiv to MAG

is different from that on MAG, we need to either add a new dense layer for class mapping or replace the last dense layer to match the output classes in MAG. Figure 10(a) shows the target-task test accuracy of 3-layer GLTs (or GNNs) where the last dense layer is replaced. We can see that the post-trained GLT and post-trained GNN identified on the source task have a similar transfer ratio on the target task. Both the post-trained GLT and post-trained GNN have significant improvement over the randomly initialized GNNs. Also, we can observe improvement of re-initialized GLT over randomly initialized GLT. As shown in Figure 10(b), similar results can be observed when adding a new dense layer. Lastly, Figure 10(c) investigates the relation between sparsity and transfer ratio (recall that we only transfer WS rather than GS), where we can see that the performance of the target task decreases as the sparsity of GLT increases.

