# OpenReview forum: "Rethinking Graph Lottery Tickets: Graph Sparsity Matters"
_ICLR.cc/2023/Conference — ICLR 2023 poster_

### Official Review · Reviewer_75Eg · 2022-10-20

**Confidence:** 4
**Correctness:** 3
**Technical Novelty And Significance:** 3
**Empirical Novelty And Significance:** 3
**Recommendation:** 8

**Clarity, Quality, Novelty And Reproducibility:**

The paper is well written and the experiments are well described. The authors propose a significant improvement of a state-of-the-art method.

**Strength And Weaknesses:**

Strengths:
- the paper is well written and easy to follow
- the proposed method effectively solves one of the main weaknesses of UGS
- the proposed method significantly improves the performance of UGS
Weaknesses:
- the experiments that assess the performance of the proposed sparsification method are conducted on three datasets: Cora, Citeseer and Pubmed. Even though these datasets are quite popular in this field, it is well known that they are not very reliable (see, e.g., the discussion in [*]). For this reason, it is better to evaluate the performance on other datasets such as OGB datasets or [*] in order to fairly assess the performance of the proposed method.
- the analysis on the transferability of graph lottery tickets seems a bit disconnected from the rest of the paper. If I understood the paper correctly, the main goal of the paper is to propose a method for improving the performance of UGS on sparse graphs. The focus of this part is thus on the sparsity of the graph. Instead in Sec. 4 the authors analyze the transferability of the sparsified subnetwork to others graphs. Then, the focus in this case is on the sparsity of the network. For this reason, I cannot see the connection between this section and the rest of the paper. In addition, it doesn't seem to me that the proposed method can provide some advantages with respect to the transferability of the lottery ticket.

[*] Dwivedi et al., Benchmarking Graph Neural Networks

**Summary Of The Paper:**

This paper extends UGS in order to improve its performance when the graph sparsity is high. The authors observe that the performance of UGS drops when the graph sparsity goes beyond a certain extent. The authors explain that this is due to the fact that only a small portion of the elements in the adjacency matrix are involved in the loss calculation. To address this problem, the authors introduce a new loss term to take into account all the elements of the adjacency matrix. They also improve the robustness of lottery tickets with sparse graphs by defining graph pruning as an adversarial graph perturbation and formulating it as a min-max optimization problem.

**Summary Of The Review:**

I think that this paper could be a solid work if the experimental evaluation is improved by adding some experiments on more reliable datasets (see comments above).

---

> ### Author Response · Authors · 2022-11-12
> **Thanks a lot for the encouraging and helpful comments! We respond to the last two comments below.**
>
> === Regarding evaluation with the OGB datasets
>
> We have, in fact, evaluated the transferability of our GLT on two OGB datasets in Figure 10: arXiv and MAG.
> We add the comparison between UGS and our method on arXiv and MAG in the revision, similar to those in Figures 5-7.
> The rebuttal revision with new experiments will be uploaded before the rebuttal deadline.
>
>
> === Regarding the GLT transferability part
>
> Since our work is the only work after UGS to study the problem of GLT, we would like to cover various facets of GLT sparsification. Section 3 is to show that our two techniques can improve the robustness of GLT even when sparsification is high, while Section 4 is to show that our weight sparsification can transfer to a new graph.
>
> The goal of the transferability section (Section 4) is not to show that our sparsification method is more advantageous than UGS.
>
> Most existing papers on LTH just find the lottery ticket but do not show how the winning ticket can be used in real applications. Therefore, with the latter part on GNN transfer learning, our goal is to revisit the motivation of lottery tickets in our GNN context and to demonstrate a practical use of our GLT to inspire future works on LTH applications.
>
> Before LTH, a sub-network found by prior sparsification methods is not retrainable. Even though "retrainability" is at the core of LTH, we are asking the question: why do we need to retrain the sub-network when we already find the sparsified model? Here, we answer this question in the context of GNN, by showing that our GLT is not only retrainable on the original graph data, but can also transfer to a new graph data which inherits the weight sparsification of the winning ticket.

---

> > ### Author Response · Authors · 2022-11-19
> > **Result on the OGB datasets**
> >
> > As suggested by the reviewer, we have added new experiments to assess the performance on two OGB datasets: arXiv and MAG.
> > In Table 5 (new version), we can observe that our method can result in higher accuracy than UGS with comparable
> > graph sparsity and weight sparsity after 5, 15, and 20 iterations.

---

> > > ### Comment · Reviewer_75Eg · 2022-12-13
> > > **Response to the authors**
> > >
> > > Thanks for your responses, the authors have addressed my comments. I will slightly raise my grade.

---

> > > > ### Author Response · Authors · 2022-12-13
> > > > **Thanks**
> > > >
> > > > We sincerely thank the reviewer for taking the time to review our response.

---

### Official Review · Reviewer_DAbQ · 2022-10-24

**Confidence:** 4
**Correctness:** 3
**Technical Novelty And Significance:** 3
**Empirical Novelty And Significance:** 2
**Recommendation:** 6

**Clarity, Quality, Novelty And Reproducibility:**

The paper is mostly well-written and easy to follow. The novelty of adversarial perturbation is novel in the context of LTH and UGS. No code to support reproducibility.

**Strength And Weaknesses:**

Strength:
1. This paper studies a practical and challenging topic in GNNs. As the scale of graph data becomes larger, it will indeed arouse our attention to graph pruning.
2. This work formulates the pruning process as a min-max optimization problem to gain the robustness of lottery tickets when the graph sparsity is high, which is novel to me.
3. This paper is easy to follow, with clear motivation and questions to address.

Weaknesses:
1. Firstly, I am extremely confused about the relationship between the problem caused by the excessive graph sparseness and the GNN transfer learning problem studied in this paper. The latter one (graph transfer learning) seems an independent part. Moreover, regarding transferring winning tickets, it seems that [1] has already done it.
2. The loss function in UGS is $L_{UGS} = L({m_g⊙A,X},m_θ⊙Θ)+r_1 ||m_g ||_1+r_2 ||m_θ ||_1$. Note that there are regularization terms in the loss function to control model sparsity and graph sparsity, but this paper only has the cross-entropy loss. It seems inconsistent and unfair to compare your model with UGS.
3. The paper uses too much vspace, resulting in an unfriendly layout for the audience. a) The formula of the “graph sparsity” and “weight sparsity” take up too much space. b) The layout of Figure 1 and 2 can also be adjusted accordingly (e.g., combining them). c) The whole algorithm process is a bit laborious for me to read, especially how to perform the network training process. An algorithm table may help.
4. The paper separates the potential nodes of class c and the other nodes by Eq. (3), is the node number of $c$ identical to $\bar{c}$? Can the Wasserstein distance measure the distance between two different node number distributions?
5. The authors only evaluate the proposed method on 2-layer GNNs, which is not universal and convincing. For some deep GNNs, do we obtain similar observations to Figure 1? In my view, a very deep GNN can pass gradients to most of the edges.
6. The improvement of the proposed method over UGS is not significant as shown in the experiments. I believe this work still has space to improve.

I may have some misunderstanding above, please feel free to correct it in the rebuttal phase. Thank you.

====after rebuttal====
W1-W4 have been addressed by the authors. I still have concerns with W5 and W6.




[1] Morcos A, Yu H, Paganini M, et al. One ticket to win them all: generalizing lottery ticket initializations across datasets and optimizers[J]. Advances in neural information processing systems, 2019, 32.


**Summary Of The Paper:**

This paper studies the pruning problem of graph neural networks (together with the adjacency matrix). It claims that the GNN performance drops sharply when the graph sparsity is beyond a certain extent in UGS. To address this issue, this work adopts two approaches: (1) adding a new auxiliary loss based on WD to better guide the edge pruning by involving the entire adjacency matrix. (2) regarding unfavorable graph sparsification as an adversarial data perturbation and formulating the pruning process as a min-max optimization problem. Further, this paper investigates the obtained winning tickets under a transfer learning setting.

**Summary Of The Review:**

This paper addresses a challenging and practical problem. Although this paper has some merits, I would like to recommend "borderline rejection" before the rebuttal phase given the weaknesses listed above.

---

> ### Author Response · Authors · 2022-11-12
> **Thank you for the insightful and constructive comments. We address the comments below.**
>
> === Weakness 1:
>
> Most existing papers on LTH just find the lottery ticket but do not show how the winning ticket can be used in real applications. Therefore, with the latter part on GNN transfer learning, our goal is to revisit the motivation of lottery tickets in our GNN context and to demonstrate a practical use of our GLT to inspire future works on LTH applications.
>
> Before LTH, a sub-network found by prior sparsification methods is not retrainable. Even though "retrainability" is at the core of LTH, we are asking the question: why do we need to retrain the sub-network when we already find the sparsified model? Here, we answer this question in the context of GNN, by showing that our GLT is not only retrainable on the original graph data, but can also transfer to a new graph data which inherits the weight sparsification of the winning ticket.
>
> Thanks for referring us to [1]! The work also studies transferring winning tickets, but its context is convolutional neural networks
>  (VGG19 and ResNet-50) which is different from our target models, i.e., GNN, which additionally introduces the problem of graph sparsification in addition to weight sparsification. We will cite [1] and include the above discussion in the revision.
>
> === Weakness 2:
>
> Thanks for pointing out the regularization terms! However, their actual code does not use this regularization. See https://github.com/VITA-Group/Unified-LTH-GNN/blob/main/NodeClassification/main_pruning_imp.py#L83 where only cross-entropy loss is used. We implement our model by referring to the source code of UGS and adding our new formulation, so we follow their loss design without regularization. In fact, if we use the regularization formulation in their paper, the performance of UGS will even drop. We will further confirm the loss inconsistency issue between their paper and their code by emailing the authors of UGS.
>
> === Weakness 3:
>
> Thanks a lot for the advice! We will improve our writing accordingly.
>
> === Weakness 4:
>
> Wasserstein distance can be used to measure the distance between two distributions with different node numbers, which should be clear from the following paper:
>
> Aaditya Ramdas, Nicolás García Trillos, Marco Cuturi: On Wasserstein Two-Sample Testing and Related Families of Nonparametric Tests. Entropy 19(2): 47 (2017)
> https://arxiv.org/abs/1509.02237
>
> Specifically, the beginning of Section 2 says "More formally, given i.i.d. samples X_1, ..., X_n ∼ P and Y_1, ..., Y_m ∼ Q, where P and Q are probability measures on R^d". Note that m does not necessarily equal to n.
>
> As another example, please refer to the Wasserstein distance function in Scipy library https://www.programcreek.com/python/example/115685/scipy.stats.wasserstein_distance. In the 2nd and 3rd examples there, the value of m does not equal to n.
>
> === Weakness 5:
>
> We agree that a very deep GNN can pass a gradient to most of the edges. However, we seldom use GNNs of more than 2 or 3 layers in practice due to the oversmoothing issue intrinsic to GNNs. GNN oversmoothing has been a hot research topic with many works, see, for example, the Dropedge paper below which indicates that the performance will drop significantly when the number of layers is larger than 2 (see their Figure 3).
>
> Yu Rong, Wenbing Huang, Tingyang Xu, Junzhou Huang: DropEdge: Towards Deep Graph Convolutional Networks on Node Classification. ICLR 2020.
> https://arxiv.org/abs/1907.10903
>
> === Weakness 6:
>
> We understand that in some subfigures in Figures 5-7, the green line (our method) may not appear significantly higher than the yellow line (UGS), but the actual number is significant. The reason that the lines are close is, for example, our y-range in Figure 5 is around [40%,80%], so for instance, 2% improvement is not giving a large visual gap. For example, with similar sparsity levels, consider the height (i.e., accuracy) difference between the red star and the blue circle on Cora in Figure 5. The improvement for GCN is 75.1% - 72.5% = 2.6%, while on Citeseer, that improvement is 70.3% - 66.7% = 3.6%.
>
> Moreover, as the sparsity becomes very high, both lines drop quickly, so they may appear close but if you draw a vertical line (i.e., in the same high sparsity level), you will see that the values of the methods are different by a large margin. Consider, for example, the height difference between the red star and the blue circle in Figures 6 and 7. Recall that we put a pink star and a cyan circle on the lines for the last iteration where the accuracy is still higher than the original model. In Figure 5, GCN on Cora, the sub-networks found by our method have a much lower sparsity than those found by UGS. For example, the graph sparsity and weight sparsity of GCN are boosted to (18.52%, 59.12%) from the (9.76%, 36.42%) of UGS.
>
> We agree that there is still room to improve. Thanks a lot for pointing this out! We will explore more sparsification techniques as a future work.

---

> > ### Author Response · Authors · 2022-11-19
> > **Follow up on the response to Weakness 3**
> >
> > As suggested by the reviewer, we have combined Figure 1 and Figure 2. An algorithm has been added on Page 5 to illustrate the training and pruning process.

---

> > ### Author Response · Authors · 2022-12-09
> > **Reminder**
> >
> > Dear Reviewer DAbQ,
> >
> > We truly appreciate this opportunity to improve our work.
> > We hope our response has clarified the questions and addressed the concerns raised in your valuable comments.
> >
> > Please kindly let us know if anything is unclear or if any post-rebuttal feedback is raised.

---

> > ### Comment · Reviewer_DAbQ · 2022-12-12
> > **Response to the authors**
> >
> > Dear authors,
> >
> > Thank you for your response. I think W1~W4 have been well addressed. For W5 and W6, I believe they may be the next directions to improve. I've also checked the comments of the other reviewers. After careful consideration, I would like to upgrade my recommendation score.
> >
> > --Reviewer

---

> > > ### Author Response · Authors · 2022-12-12
> > > **Thanks to Reviewer DAbQ**
> > >
> > > Thanks again for taking the time to review our submission and rebuttal!

---

### Official Review · Reviewer_FtWv · 2022-10-27

**Confidence:** 4
**Correctness:** 3
**Technical Novelty And Significance:** 2
**Empirical Novelty And Significance:** 2
**Recommendation:** 6

**Clarity, Quality, Novelty And Reproducibility:**

The paper has decent clarity and quality
The paper has some novelty, mainly applying two mathematical technique to improve the loss function and optimization of existing LTH.

**Strength And Weaknesses:**

Strengths:
1. The paper is easy to follow. The paper describes the disadvantage lies in existing work and then introduced there methods for improving.
2.  Experimental on multiple GNN models (GCN , GAT, and GIN) and datasets (Cora, Citeseer and PubMed, Arxiv and MAG) seems to be sufficient.
3. I like the ablation on the  auxiliary loss weights and the prune ratio combination of weight and graph.

Weekness:
1. The paper target the work UGS ( by Chen et al. ) directly, by improving the performance at a high graph sparsity. However, looking at Figure 5-7. I’m not sure if the improvement is significant enough, especially since some seems quite similar.
2. Less comparison with other existing work.

Questions:
1. How to get equation 10,11 from equation 9?
2. In Figure 4, the trend of Randomly initialized GLT and GNN is very similar.  What sparsity is the GLT? Does sparsity effect the converging speed?

**Summary Of The Paper:**

The paper focuses on improving lottery tickets on GNN. Although existing work (UGS) prunes graph and weight simultaneously,  the performance degrades significantly when graph sparsity is high.
The paper found out the problem in UGS lies in the loss formulation for pruning adjacency matrix, as it does not involve all edges for back propagation.To solve this, they add a new auxiliary loss head that measures the inter-class separateness of nodes with Wasserstein distance to better guide the edge pruning.
The paper also proposed an adversarial perturbation technique, which formulates the pruning process as a min-max optimization problem.   This helps improve the robustness of their graph lottery tickets against the graph sparsity.
Additionally, the paper explored graph transfer learning by running node classification tasks on two citation networks.
Results on three GNN models with multiple datasets show improvement over the existing LTH method.

**Summary Of The Review:**

Generally speaking,  the paper has decent quality.
My concern lies in the significance of novelty and performance gain over existing works.

---

> ### Author Response · Authors · 2022-11-12
> **Thank you for the detailed and helpful comments！We address the comments as follows.**
>
> === Weakness 1:
>
> We understand that in some subfigures in Figures 5-7, the green line (our method) may not appear significantly higher than the yellow line (UGS), but the actual number is significant. The reason that the lines are close is, for example, our y-range in Figure 5 is around [40%,80%], so for instance, 2% improvement is not giving a large visual gap.
>
> Moreover, as the sparsity becomes very high, both lines drop quickly, so they may appear close but if you draw a vertical line (i.e., in the same high sparsity level), you will see that the values of the methods are different by a large margin. Consider, for example, the height difference between the red star and the blue circle in Figures 6 and 7.
>
> Table 1: Cora after 17 iterations
>
> Model || GCN || GIN || GAT ||
>
> Metrics || GS(%)  | WS(%) | ACC(%) || GS(%) | WS(%) | ACC(%) || GS(%) | WS(%) | ACC(%) ||
>
> UGS || 56.28 | 97.52 | 72.5 || 57.62 | 97.25 | 69.6 || 57.42 | 97.38 | 78.7 ||
>
> Our || 58.02 | 97.80 | 75.1 || 57.88 | 98.26 | 72.8 | 58.12 | 97.89 | 80.0 ||
>
> Recall that we put a red star and a blue circle on the lines at iteration 17 so that they have similar sparsity levels.
>
> To demonstrate how significantly our method improves over UGS when sparsity level is similar, we show the height difference between the red star and the blue circle for Figure 5 using the table above. We can see that the sub-networks found by our method have much higher accuracy than those found by UGS (often 2%-3%). For example, the improvement for GCN is 75.1% - 72.5% = 2.6%, while on Citeseer, that improvement is 70.3% - 66.7% = 3.6%.
>
> Table 2: Last GLT w/ accuracy >= original model on Cora
>
> Model || GCN || GIN || GAT ||
>
> Metrics || GS(%) | WS(%) | ACC(%) || GS(%) | WS(%) | ACC(%) || GS(%) | WS(%) | ACC(%) |
>
> UGS || 9.76 | 36.42 | 81.3 || – | –  | – || 43.96 | 91.88 | 80.1 ||
>
> Our || 18.52 | 59.12 | 82.2 || 4.86 | 20.07 | 79.6 || 63.82 | 98.93 | 80.7 ||
>
> Note that as we sparsify a GCN, the accuracy often rises first and then drops. Recall that we put a pink star and a cyan circle on the lines for the last iteration where the accuracy is still higher than the original model.
>
> To demonstrate how significantly our method improves over UGS when accuracy level is similar, we show the horizontal difference between the pink star and the cyan circle for Figure 5 using the table above. We can see that the sub-networks found by our method have a much lower sparsity than those found by UGS. For example, the graph sparsity and weight sparsity of GCN are boosted to (18.52%, 59.12%) from the (9.76%, 36.42%) of UGS. This verifies that our method can improve the robustness of GLT against sparsity.
>
> === Weakness 2:
>
> GLT (graph lottery tickets) is a very new topic and we are only aware of UGS (by Chen et al.) as the only published work that studies GLT. We would be grateful if the you can point us to the other related works on GLT that you are aware of, and we will do our best to review and compare with those works before the rebuttal deadline.
>
> === Question 1:
>
> We remark that Eqs (9), (10), (11) are the update steps in each iteration to optimize the min-max objective shown in Eq (8).
>
> Eq (8) has an inner maximization on variable m_g, which we use gradient ascent to update m_g by adding \eta_1 * gradient of m_g.
>
> Once the new m_g is calculated, we proceed to the outer minimization, which is on variables m_\theta and \Theta. Here, we use gradient descent to update them by deducting \eta_2 * gradient, as shown in Eq (10) and Eq (11).
>
> Note that the gradients in Eq (10) and Eq (11) have two terms since the new m_g itself is a function of (m_\theta, \Theta) as specified by Eq (9), while L is a function of 3 variables (m_\theta, \Theta, m_g). As a result, for Eq (11) for example, (i) one term is (d L_{m_g})/(d \Theta), and (ii) the second term is based on the chain rule that times (d L_{m_g})/(d m_g) with (d m_g)/(d m_\Theta).
>
> In other words, the only connection between Eq (9) and Eqs (10)-(11) is that, given the computed m_g by Eq (9), the chain-rule path L_{m_g} -> m_g -> m_\theta or \Theta is used to derive gradient in Eq (10) and Eq (11).
>
> === Question 2:
>
> We remark that the key for GLT or a lottery ticket to converge to a higher accuracy is to use the original weight initialization to reinitialize each iteration of sparsification, and random reinitialize does not work. This is exactly the key observation and conclusion of the LTH paper (Frankle and Carbin, 2019). Here, we are just echoing this conclusion in the graph convolution setting. Note that LTH was originally only verified for the fully-connected and ConvNet settings. It is no surprise that the yellow and black lines converge to a lower accuracy in Figure 4, and the important thing is that the green line converges to a higher accuracy.
>
> Regarding the sparsity is the GLT, the graph sparsity and weight sparsity are 91.81% and 43.10%, respectively.
>
> Regarding the effect of sparsity on converging speed, please refer to Figure 10(c).

---

> > ### Comment · Reviewer_FtWv · 2022-12-08
> > **Response to authors**
> >
> > The authors addressed my questions. I will raise the score to borderline accept.

---

> > > ### Author Response · Authors · 2022-12-09
> > > **Thanks**
> > >
> > > We are most grateful for all feedback you gave to us.
> > > Thanks again for taking the time to review the rebuttal.

---

### Decision · Program_Chairs · 2023-01-20

**Decision:**

Accept: poster

**Justification For Why Not Higher Score:**

This paper is potentially a foundational work for a specific type of networks (GNNs) in a specific area (lottery tickets) of a niche area (neural network pruning). It's not of general appeal to the broader community, and it's relatively less significant by virtue of its niche appeal.

**Justification For Why Not Lower Score:**

It's a solid piece of technical work that I expect will be a foundation for any further research in lottery tickets for GNNs. It improves on some relatively primitive prior work, and I expect it to be influential in this area (insofar as this area is of interest). It's a pretty niche area, though, so the impact will be limited by that.

**Metareview: Summary, Strengths And Weaknesses:**

**Summary:** This paper studies lottery tickets for graph neural networks (GNNs). This paper attempts to cover an enormous amount of ground. First, it proposes a new method for pruning GNNs that improves on a previous baseline by separately addressing the graph structure and the network weights. It then proceeds to investigate the transferability of these subnetworks, a topic that has (in other parts of the LTH literature) required entirely independent papers to take on.

**Strengths:**
* It works. The results are a marked improvement over the sole prior work (UGS). Transfer also seems to work quite well. The authors look at what I understand to be a pretty wide variety of graph tasks in doing so.
* The authors look at transfer, which is one of the ways that lottery ticket work can be made useful even through it requires training the network before pruning it.

**Weaknesses:**
* The method is somewhat complicated. In the standard lottery ticket literature, all that is needed is magnitude pruning (although fancier methods exist that give slightly better results). Here, the method requires an auxiliary loss function and pruning is formulated as a minimax problem.

**Overall:** This result seems to set a nice foundation for future work on GNN pruning. The reviewers were all in favor of acceptance, and I am as well.

**Note From Pc:**

if the above contains the word "oral" or "spotlight" please see: "oral" presentation means -> notable-top-5% and "spotlight" means -> notable-top-25%. As stated in our emails, we are disassociating presentation type from AC recommendations

**Summary Of Ac-Reviewer Meeting:**

N/A